# Repurposing of the enhancer-promoter communication underlies the compensation of *Mesp2* by *Mesp1*

Hajime Okada [1], Yumiko Saga [1,2,3] *

**1** Department of Gene Function and Phenomics, National Institute of Genetics, Mishima, Japan, **2** Department of Genetics, School of Life Science, The Graduate University for Advised Studies (SOKENDAI), Mishima, Japan, **3** Department of Biological Sciences, Graduate School of Science, The University of Tokyo, Tokyo, Japan

* ysaga@nig.ac.jp

## Abstract

Organisms are inherently equipped with buffering systems against genetic perturbations. Genetic compensation, the compensatory response by upregulating another gene or genes, is one such buffering mechanism. Recently, a well-conserved compensatory mechanism was proposed: transcriptional adaptation of homologs under the nonsense-mediated mRNA decay pathways. However, this model cannot explain the onset of all compensatory events. We report a novel genetic compensation mechanism operating over the *Mesp* gene locus. *Mesp1* and *Mesp2* are paralogs located adjacently in the genome. *Mesp2* loss is partially rescued by *Mesp1* upregulation in the presomitic mesoderm (PSM). Using a cultured PSM induction system, we reproduced the compensatory response *in vitro* and found that the *Mesp2*-enhancer is required to promote *Mesp1*. We revealed that the *Mesp2*-enhancer directly interacts with the *Mesp1* promoter, thereby upregulating *Mesp1* expression upon the loss of *Mesp2*. Of note, this interaction is established by genomic arrangement upon PSM development independently of *Mesp2* disruption. We propose that the repurposing of this established enhancer-promoter communication is the mechanism underlying this compensatory response for the upregulation of the adjacent gene.

## Author summary

Genetic compensation, the compensatory response by upregulating another gene or genes, is one of the inherent mechanisms against gene disruption to confer cellular fitness. However, the regulatory mechanisms are largely unknown. Nonsense-mediated mutant mRNA degradation was recently proposed as a conserved mechanism across species to upregulate homologous genes to compensate for a disrupted gene, but this cannot explain compensation events with no mutant mRNA. This study investigated the compensation mechanism operating over adjacent paralogs, *Mesp1* and *Mesp2*, in the genome. *Mesp* genes encode essential transcription factors in the presomitic mesoderm for development. In general, an enhancer is considered to activate a target gene when it physically interacts

**Data Availability Statement:** All relevant data are within the manuscript and its Supporting Information files.

**Funding:** This work was supported by NIG Postdoctoral Research Fellow grant 2018 (to HI)

and the Japan Society for the Promotion of Science KAKENHI grants 19K16152 (to HI). The funders had no role in study design, data collection and analysis, decision to publish, or preparation of the manuscript.

**Competing interests:** The authors have declared that no competing interests exist.

with the target. The communication of the *Mesp2*-enhancer with the *Mesp1* promoter is established upon differentiation of the presomitic mesoderm, but this communication activates *Mesp1* only when *Mesp2* is disrupted, leading to compensation. We revealed a novel compensation mechanism depending on the repurposing of this enhancer-promoter communication by gene disruption. Our study also provides new insight into transcriptional regulation by providing the concept that an enhancer changes its target even among its physically interacting genes in a context-dependent manner.

## Introduction

Organisms, especially multicellular organisms, are programmed to develop tissues, organs, and whole bodies. In parallel with these programs, buffering systems against gene network perturbations are inherently present [1]. Genetic compensation is one of these buffering systems and is considered to make up for network failure. Although compensatory events in several model organisms have been extensively documented since yeast in 1969 [2], the underlying mechanisms have been described as a consequence of the loss of protein function [2,3] or simply as unknown. Recently, a nonsense-mediated mRNA decay (NMD) pathway-mediated transcriptional adaptation model was proposed. In this model, mutant mRNA bearing a pretermination codon (PTC) is degraded, and the subsequent pathways modify and activate the chromatin state of the homologous genes of the mutant via fragments of the degraded RNA [4–7]. Thus, this mechanism explains the upregulation of homologous genes independent of the downstream protein loss.

Although several genetic compensation events induced by genetic mutations can be explained by the NMD-mediated model [5,6], there are exceptional compensatory events. One example is observed in the mutation of mouse *Mesp2*, encoding a transcription factor required for somitogenesis. *Mesp* genes, *Mesp1* and *Mesp2*, are located in a head-to-head orientation on chromosome 7 (Fig 1A), and these genes are co-expressed in the nascent mesoderm and PSM. Replacement of the *Mesp2* coding region with *Mesp1* almost completely rescues the *Mesp2* defect [8], suggesting that the functions of MESP proteins are almost identical. Targeted disruption of *Mesp2* induces the compensatory upregulation of *Mesp1*, which partially ameliorates the *Mesp2* defect in the PSM [9]. Of note, *Mesp2*-knockout (KO) mice were generated by replacing endogenous genes with exogenous genes, such as *MerCreMer* [9]; thus, these mutants do not contain a mutant mRNA-bearing PTC. This strongly suggests that this compensatory response does not rely on the NMD-mediated transcriptional adaptation, but instead on the downstream of MESP2 loss or other compensatory mechanisms.

Thus far, enhancer elements have been identified in the intergenic DNA region between *Mesp1* and *Mesp2* [10,11]. The intergenic enhancer (I-enhancer) located 4k bp distal from the *Mesp1* promoter was assigned as a *Mesp1*-enhancer, and the proximal enhancer to the *Mesp2* promoter (P2-enhancer) was assigned as a *Mesp2*-enhancer based on reporter mouse assays using a candidate DNA element linked with reporter gene [10]. Further bacterial artificial chromosome (BAC) transgenes, which contain 160k bp spanning the genomic context, including the *Mesp* gene locus [11], demonstrated the necessity of these elements for each gene activation. However, although these assays determined the individual enhancer elements, even BAC transgenic reporter assays did not necessarily recapitulate the redundant regulation by enhancers in the genuine genomic context [12]. Moreover, both *Mesp1* and *Mesp2* are simultaneously expressed in the same regions of the nascent mesoderm and the PSM [13], which makes it more difficult to address whether these enhancers regulate each *Mesp* gene in a one-

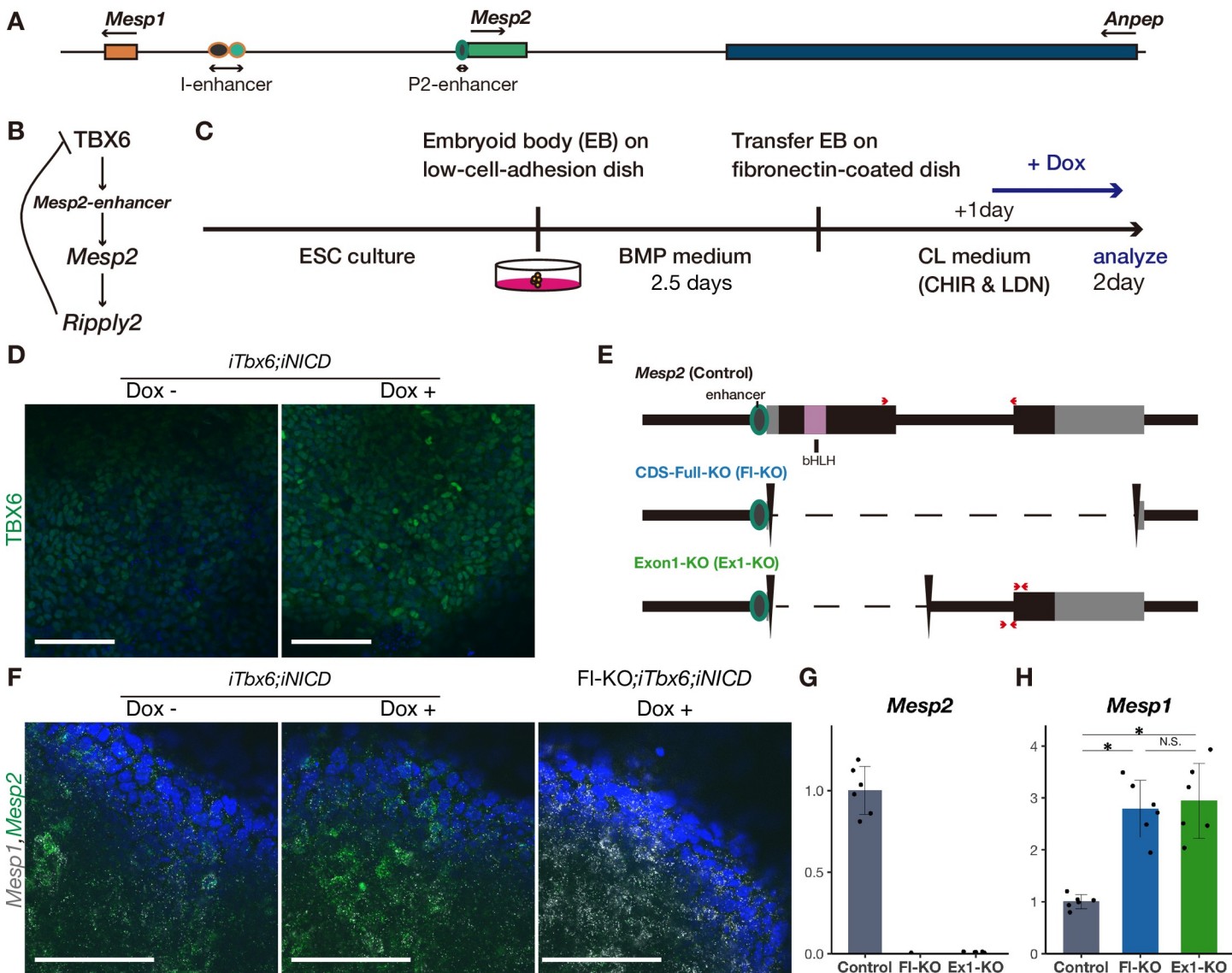

**Fig 1. Reproducing the PSM and compensatory response of *Mesp* genes *in vitro*.** A Schematic diagram of the genome at the *Mesp2* locus and enhancers. B Schematic diagram of reciprocal regulation of *Mesp2*. C Schedule of *in vitro* PSM induction from ES cells for optimal induction of *Mesp2*. D Immunostaining of TBX6 in the *in vitro* PSM of *iTbx6;iNICD* with (right) or without (left) Dox administration. Scale bars: 100 μm. E Schematic diagram of *Mesp2* coding sequence deletions. Red arrows indicate the primers used for qPCR. Primers at *Mesp2* (wild-type), which detect only spliced *Mesp2*, were used for most experiments. Primers at Exon1-KO were used for precursor RNA and either precursor or mature RNA of *Mesp2*, as used in Fig 2A. F *In situ* hybridization for *Mesp1* (white) and *Mesp2* (green) in the *in vitro* PSM of *iTbx6;iNICD* without (left) or with (middle) Dox administration, and *iTbx6;iNICD;Mesp2*-CDS-Full-KO upon Dox administration (right). Scale bars: 100 μm. G, H qPCR analysis of *Mesp2* (G) and *Mesp1* (H) in the *in vitro* PSM (n = 6 cultures for each genotype). The expression levels are presented as the ratio against *iTbx6;iNICD* line (control). P-values were calculated by the Mann-Whitney U test comparing *iTbx6;iNICD* (control) and *iTbx6;iNICD;Mesp2*-KO lines. Data are presented as the mean ± SD. *Asterisk* indicates significant (p < 0.05).

to-one or a multi-target manner. The activation of either enhancer requires their T-box motifs during somitogenesis [11,14]. Somites are sequentially formed from the PSM in the anterior to posterior direction. *Mesp1* and *Mesp2* are cyclically upregulated via collaborative Notch signaling and TBX6 in the expected next somite boundary [13,15,16]. TBX6 is a T-box transcription factor that binds to either the I-enhancer [17] or the P2-enhancer [14,18]. MESP2 directly induces *Ripply2* [19], and RIPPLY2 then degrades TBX6 through a proteasome pathway leading to the reciprocal transcriptional repression of *Mesp2*, which assures the temporal activation

and termination of the *Mesp2* gene (Fig 1B) [16,20,21]. When *Mesp2* is disrupted, TBX6 is not sufficiently degraded due to the absence of RIPPLY2 [20,22] and may upregulate *Mesp1* via the I-enhancer. Thus, MESP2 protein loss may affect *Mesp1* upstream-regulation, and this break-down of the negative feedback loop is a possible mechanism of this genetic compensation. In this study, we investigated this possibility and explored a new compensatory mechanism for *Mesp* genes independent of the downstream of MESP2 loss.

## Results

### Establishing an *in vitro* PSM induction system that reproduces the compensatory response of *Mesp1* for *Mesp2* loss

To explore a novel compensatory mechanism for *Mesp* genes during somitogenesis, we took advantage of an established *in vitro* PSM induction system based on the reported protocol, which mimics the developmental process [23]. This protocol involves the creation of embry-oids (EBs) of ES cells in BMP-containing medium and subsequent culture with a BMP inhibitor (LDN193189) and Wnt agonist (CHIRON99021) to induce PSM efficiently (Fig 1C). However, we were unable to consistently induce *Mesp2* in this system even though we induced PSM differentiation using the same wild-type (WT) ES cells (S1A Fig). The failed induction of *Mesp2* corresponded to lower *Tbx6* and *Notch1* induction, which are essential factors for *Mesp2* induction [14] (S1A Fig). To reproducibly induce *Mesp2*, we established an ES cell line containing inducible *Tbx6* and *NICD*, Notch1 intracellular domain which translocates into the nucleus and interacts with transcriptional regulators, upon doxycycline (Dox) administration, termed the inducible *Tbx6* and *NICD* (*iTbx6;iNICD*) line. Using this cell line, we induced PSM differentiation with minor optimization of the reported protocol (Fig 1C and see *In vitro* PSM induction in Materials and Methods). Immunocytochemistry (ICC) analysis confirmed the reproducible induction of TBX6 (Fig 1D). We also confirmed the induction of essential developmental genes along with the PSM differentiation process (S1B and S1C Fig). Early meso-derm markers, *Brachyury* (*T*) and *Eomes*, were downregulated, and the segmentation clock gene *Hes7*, and segmentation boundary genes *Mesp2* and *Ripply2* were induced by Dox-induced TBX6 and NICD (S1C Fig). This gene expression pattern supports the efficient induction of PSM in this system.

Next, to examine whether the compensatory response occurred in the cultured system, we deleted the genomic region of *Mesp2* in the *iTbx6;iNICD* ES cell line (the scheme is shown in Fig 1E). Using two coding sequence (CDS)-deleted *Mesp2*-KO lines, CDS-Full-KO (Fl-KO) and Exon1-KO (Ex1-KO) ES cells, we induced PSM differentiation and analyzed the expression level of *Mesp* genes (Fig 1F–1H). As previously observed in *Mesp2*-KO mice [9], *Mesp1* was upregulated in the *in vitro* PSM using these two KO lines to an almost equivalent level (Fig 1H). Thus, we were able to reproduce the compensatory response in the *in vitro* PSM. Hereafter, we refer to the *iTbx6;iNICD* line and derivative *Mesp2*-KO lines as control and *Mesp2*-KO, respectively.

### The P2-enhancer is required for this compensatory response independent of the NMD pathway

The NMD pathway requires mutant mRNA-bearing PTC in the exons before the last exon during the precursor mRNA (pre-mRNA) splicing [24]. Considering this rule, it is likely that NMD pathways are not activated in *Mesp2*-KO lines because they are lacking almost the entire exon 1 and have no or one exon left (Fig 1D). To clarify this hypothesis that the NMD pathway was not activated in the *Mesp2*-KO lines, we investigated the lack of mRNA degradation after

splicing. We examined the expression of pre-mRNA and mature mRNA in the Exon1-KO line, which retains the posterior intronic element and exon 2, using qPCR (primers are shown in Fig 1D). The expression levels of these *Mesp2* transcripts were markedly reduced in this KO line (Fig 2A). This suggests that *Mesp2*-locus transcription itself was suppressed in the Exon1-KO, raising the possibility that this compensatory response occurred independently of the NMD pathway stimulated by PTC-bearing mRNA.

The above results led us to hypothesize that the absence of *Mesp2* transcription triggers the compensatory response. To test this hypothesis, we deleted the entire *Mesp2* gene, including both the enhancer and CDS regions, termed En-CDS-KO (Fig 2B). *Mesp2* was depleted in this line, but *Mesp1* was not upregulated (Fig 2C). Thus, the absence of *Mesp2* transcription is not the trigger of the compensatory response and raised another possibility that the P2-enhancer is required for this compensation. To test this possibility, we deleted only the P2-enhancer region, termed Enhancer-KO (En-KO) (Fig 2B). *Mesp2* transcription was absent in this mutant line, as expected. Consistent with the En-CDS-KO line, *Mesp1* was not upregulated in the Enhancer-KO line (Fig 2D), indicating that the P2-enhancer is required for this compensation.

The P2-enhancer has four TBX6 binding sites (Fig 2E), and multiple mutations in these TBX6-binding sites reduce the activity of P2-enhancer in a stepwise manner [18]. To address whether activation of the P2-enhancer is required for this compensatory response, we disrupted three or four TBX6-binding sites by introducing nucleotide substitutions as reported previously [14,18] in the *iTbx6;iNICD* line, referred to as EmT3 and EmT4 lines, respectively (Fig 2E). We induced the differentiation of EmT3 and EmT4 lines into PSM, and analyzed the expression of *Mesp* genes. As expected, the expression of *Mesp2* was markedly suppressed (Fig 2F) and compensatory upregulation of *Mesp1* did not occur (Fig 2G). Collectively, this indicated that P2-enhancer activity is required for this compensation (summarized in S2 Fig).

## The compensatory response is independent of the breakdown of the negative feedback loop

The P2-enhancer is required for compensation independent of the NMD pathway. In addition to the NMD pathway, the direct effects of the loss of protein function have been reported as a primary trigger of a compensatory response [2]. For example, the disruption of *Rpl22* upregulates its paralog *Rpl22l1*, the expression of which is normally inhibited by *Rpl22* [3]. Therefore, breakdown of the negative feedback loop by gene disruption may induce the upregulation of its counterpart homologous gene. In the case of *Mesp*, MESP2 induced by TBX6 induces *Ripply2* [19], and then RIPPLY2 degrades TBX6 through a proteasome pathway [16,20,21]. Thus, in the *Mesp2*-KO mice, *Ripply2* is not sufficiently induced and TBX6 is not degraded [20,22] (Fig 3A). We thus asked whether this breakdown of the negative feedback loop by MESP2 protein loss is involved in this P2-enhancer-based compensation.

To confirm the breakdown of the negative feedback loop, we examined the expression of *Ripply2* and TBX6 in the *Mesp2*-KO ES cell lines. As expected, *Ripply2* expression was reduced (Fig 3B). However, TBX6 was not fully degraded even in the control PSM, and the protein level of TBX6 did not always increase in the *Mesp2*-KO PSM compared with the control (Figs 3C and 3D and S3A). TBX6 is degraded by RIPPLY2 at the protein level, not at the mRNA level [20,21], and indeed, the mRNA level of *Tbx6* was also comparable between the control and mutants (S3B Fig). These suggested that the negative feedback loop of TBX6-MESP2-RIPPLY2 did not function properly in our *in vitro* system, probably due to the insufficient expression of *Ripply2* during the constant induction of TBX6.

To confirm that the exogenous TBX6 was not degraded because of a lack of RIPPLY2 induction, we established *iMesp2* and *iRipply2* lines, in which *Mesp2* or *Ripply2* can be

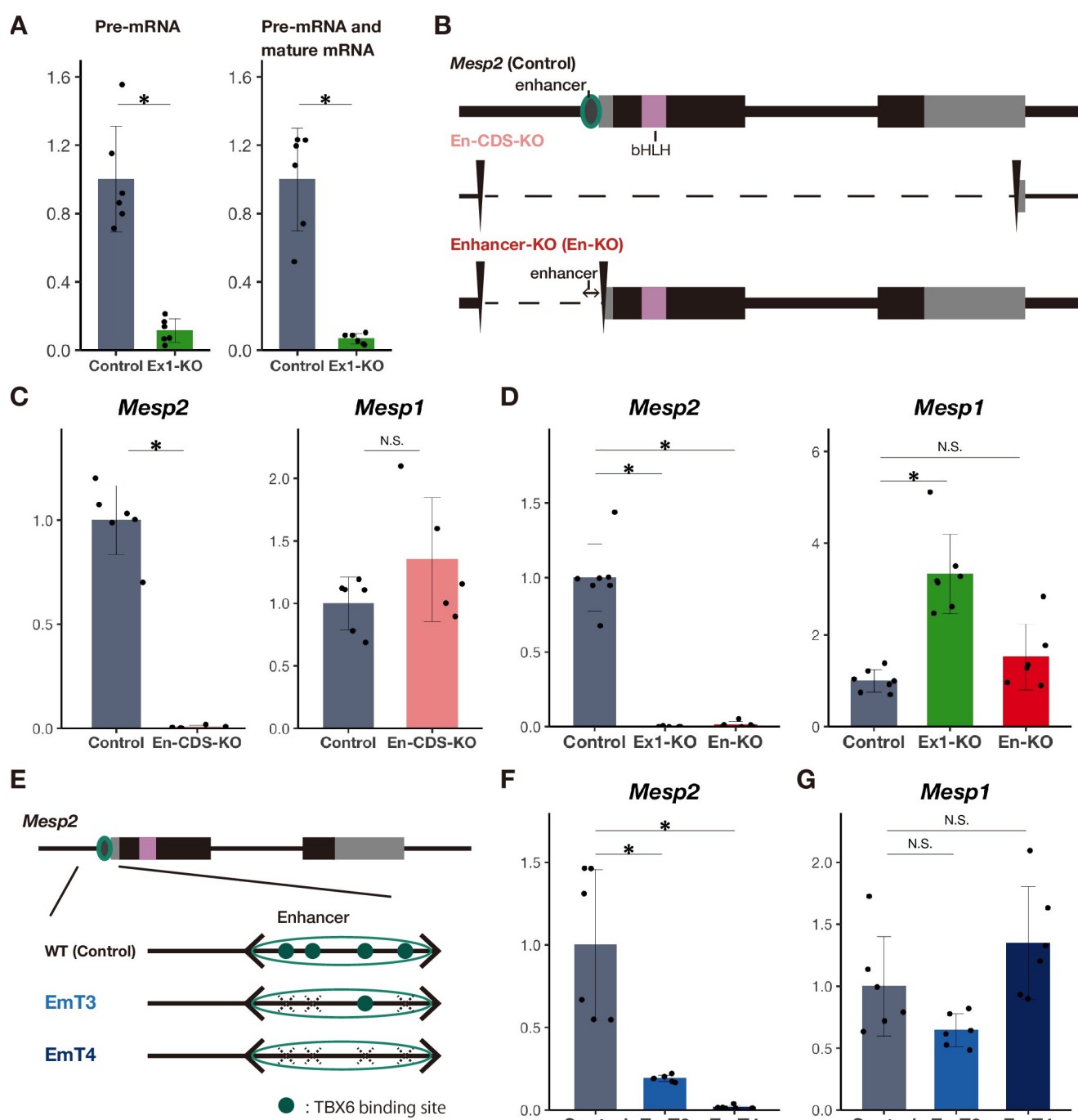

**Fig 2. The P2-enhancer is required for the compensatory response.** A qPCR analysis of precursor RNA (Pre-mRNA) and either precursor or mature RNA of *Mesp2* (n = 6 cultures for each genotype). Used primers are shown in Fig 1D. The expression levels are presented as the ratio against *iTbx6;iNICD* line (control). B Schematic diagram of *Mesp2* genomic deletions of En-CDS-KO and Enhancer-KO. C qPCR analysis of *Mesp2* and *Mesp1* in *in vitro* PSM (n = 5 or 6 cultures for each genotype). The expression levels are presented as the ratio against *iTbx6;iNICD* line (control). *P*-values were calculated by the Mann-Whitney U test comparing *iTbx6;iNICD* (control) and *iTbx6;iNICD*;P2-En-CDS-KO. *Asterisk* indicates significant ($p < 0.05$). D qPCR analysis of *Mesp2* and *Mesp1* in *in vitro* PSM (n = 6 to 7 cultures for each genotype). The expression levels are presented as the ratio against *iTbx6;iNICD* line (control). *P*-values were calculated by the Mann-Whitney U test comparing *iTbx6;iNICD* (control) and *iTbx6;iNICD*;*Mesp2*-Exon1-KO or *iTbx6;iNICD*; P2-Enhancer-KO. Data are presented as the mean ± SD. *Asterisk* indicates significant ($p < 0.05$). E Schematic diagram of mutations of T-boxes in the P2-enhancer. F, G qPCR analysis of *Mesp2* (F) and *Mesp1* (G) in *in vitro* PSM (n = 6 cultures for each genotype). The expression levels are presented as the ratio against *iTbx6;iNICD* line (control). *P*-values were calculated by the Mann-Whitney U test comparing *iTbx6;iNICD* (control) and *iTbx6;iNICD*; P2-Enhancer mutant lines. *Asterisk* indicates significant ($p < 0.05$).

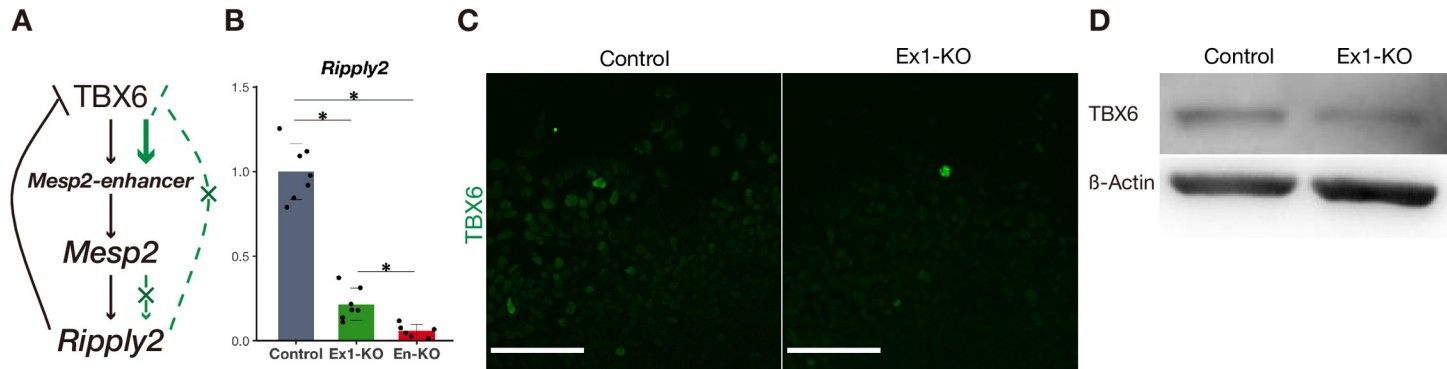

**Fig 3. The negative feedback loop does not function properly in the *in vitro* PSM system.** A Schematic diagram of reciprocal regulation of *Mesp2*. Green lines and characters indicate the process of *Mesp2* disruption. B qPCR analysis of *Ripply2* in *in vitro* PSM (n = 6 or 7 cultures for each genotype). The expression level is presented as the ratio against *iTbx6;iNICD* line (control). *P*-values were calculated by the Mann-Whitney U test comparing *iTbx6;iNICD* (control) and *iTbx6;iNICD;Mesp2*-Exon1-KO or *iTbx6;iNICD*;P2-Enhancer-KO, or comparing *iTbx6;iNICD;Mesp2*-Exon1-KO and *iTbx6;iNICD*;P2-Enhancer-KO. Data are presented as the mean ± SD. *Asterisk* indicates significant ($p < 0.05$). C Immunostaining of TBX6 in the *in vitro* PSM of *iTbx6;iNICD* (control) (left) and *iTbx6;iNICD;Mesp*2-Exon1-KO (right) upon Dox administration. Scale bars: 100 μm. D Western blotting analysis to monitor the degradation of TBX6 in the *in vitro* PSM of *iTbx6;iNICD* (control), and *iTbx6;iNICD;Mesp2*-Exon1-KO upon Dox administration. ß-Actin was used as the internal control of the protein lysate.

additionally induced upon Dox administration, leading to the simultaneous induction of *Tbx6* and *NICD* (S3C–S3E Fig). Under such conditions, TBX6 protein was degraded in the *iRipply2* line but not in the *iMesp2* line (S3F and S3G Fig). This difference can be ascribed to the different amounts of *Ripply2* induced in each line (S3E Fig), indicating that that the dynamic range of the negative feedback loop of TBX6-MESP2-RIPPLY2 in our *in vitro* system is narrow and only effective when *Ripply2* is directly induced exogenously. Thus, collectively, genetic compensation operating over *Mesp* genes is independent of the breakdown of the negative feedback loop in our *in vitro* system.

## The P2-enhancer promotes *Mesp1* expression in cooperation with the I-enhancer

Our study revealed that the P2-enhancer plays a central role in compensation. The next question is how the P2-enhancer regulates this compensation. Enhancers are DNA regulatory elements of gene transcription [25]. However, the P2-enhancer is not necessary to activate *Mesp1*, at least under physiological conditions, because the deletion or inactivation of this enhancer did not lead to the reduction of *Mesp1* (Fig 2C, 2D and 2G). If the P2-enhancer promotes the expression of *Mesp1* upon the deletion of the *Mesp2* coding sequence, this enhancer should alter its target from *Mesp2* to *Mesp1*. To test whether the P2-enhancer selectively upregulates *Mesp1* or simply affects proximal gene expression, we examined the other gene proximal to *Mesp2*, *Anpep*, which is located adjacent to *Mesp2* on the opposite side of *Mesp1* (Fig 1A). The protein encoded by *Anpep* is an alanyl aminopeptidase that localizes in the plasma membrane and digests peptides, differing in function from MESP proteins. Of note, this non-homologous gene was also upregulated in the condition where *Mesp1* was upregulated (Fig 4A). This supports the idea that the P2-enhancer simply affects the proximal genes rather than selectively affecting *Mesp1*.

In the presence of *Mesp2*, *Mesp1* must be regulated by the I-enhancer. We hypothesized that the *Mesp1* promoter interacts with the I-enhancer but not with the P2-enhancer under physiological conditions (Fig 4B). On the other hand, in the absence of *Mesp2*, *Mesp1* and *Anpep* promoters should interact with the P2-enhancer. In this situation, there are two possibilities regarding the manner by which the P2-enhancer interacts with the *Mesp1* promoter.

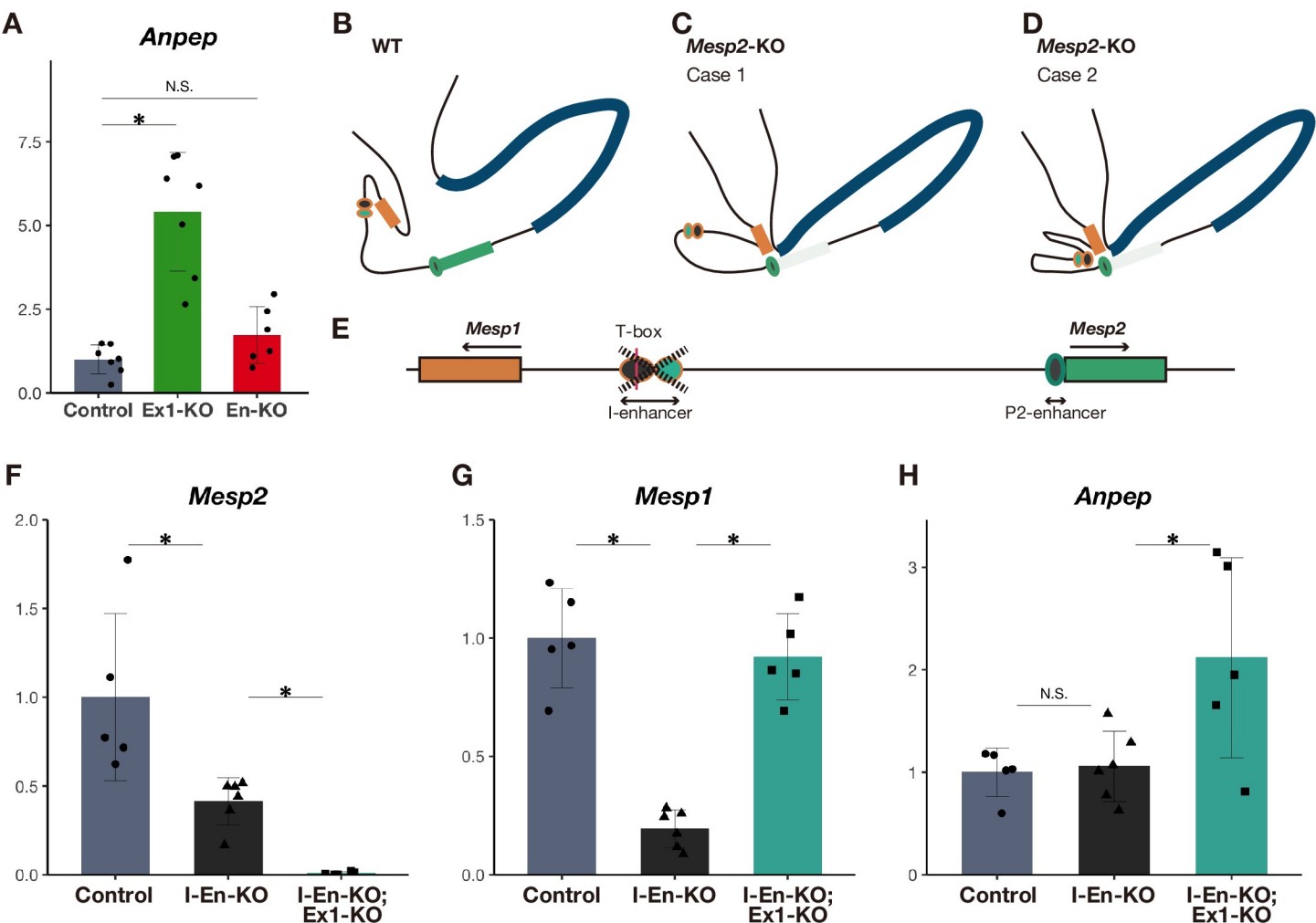

**Fig 4. The I-enhancer and P2-enhancer cooperatively promote *Mesp1* in the absence of *Mesp2*.** A qPCR analysis of *Anpep* in *in vitro* PSM (n = 6 to 7 cultures for each genotype). The expression level is presented as the ratio against *iTbx6;iNICD* line (control). *P*-values were calculated by the Mann-Whitney U test comparing *iTbx6;iNICD* (control) and *iTbx6;iNICD;Mesp2*-Exon1-KO or *iTbx6;iNICD;*P2-Enhancer-KO. *Asterisk* indicates significant ($p < 0.05$). B–D The genomic conformation models around the *Mesp2* locus in the presence (B) and absence (C and D) of the *Mesp2* gene. Orange, green, and blue rectangles indicate *Mesp1*, *Mesp2*, and *Anpep*, respectively. Circles indicate I- (orange line) and P2- (green line) enhancers. The I-enhancer interacts with the *Mesp1* promoter in the presence of *Mesp2* (B). If this enhancer interacts with the *Mesp1* promoter in the absence of *Mesp2* (shown as light green), the genome structure should be like case 2 (D). If not, it should be like case 1 (C). E Schematic diagram of genomic deletion of the I-enhancer. Orange-lined black and emerald green circles indicate the I-enhancers activated during early mesodermal and PSM stages and early mesodermal stage only [11], respectively. The red line indicates the T-box domain. F–H qPCR analysis of *Mesp2* (F), *Mesp1* (G), and *Anpep* (H) in *in vitro* PSM; *iTbx6;iNICD* (control), *iTbx6;iNICD;*I-enhancer-KO, and additive *Mesp2*-Exon1-KO or *iTbx6;iNICD;*I-enhancer-KO (n = 5 or 6 cultures for each genotype). The expression levels are presented as the ratio against *iTbx6;iNICD* line (control). *P*-values were calculated by the Mann-Whitney U test comparing *iTbx6;iNICD;*I-enhancer-KO and others. Data are presented as the mean ± SD. *Asterisk* indicates significant ($p < 0.05$).

One is that the P2-enhancer solely interacts with the *Mesp1* promoter, replacing the I-enhancer (case 1 in Fig 4C). The other is that the P2-enhancer interacts with the *Mesp1* promoter in cooperation with the I-enhancer (case 2 in Fig 4D).

To test these possibilities and elucidate the necessity of the I-enhancer for this compensation, we deleted the I-enhancer region in the *iTbx6;iNICD* cell line (I-En-KO line) (Fig 4E). This deletion led to a marked reduction of *Mesp1* expression and also reduced *Mesp2* expression, whereas *Anpep* expression was not significantly reduced (Figs 4F–4H and S4E), suggesting that the I-enhancer also promotes *Mesp2* expression. To examine the involvement of the I-enhancer in the compensatory response, we further deleted the *Mesp2* coding sequence

(shown in Fig 1D) in the I-En-KO line. *Mesp1* upregulation was observed in these lines compared with the I-En-KO line (Figs 4G and S4B), although it did not reach the level observed in the *Mesp2*-KO line containing the intact I-enhancer. *Anpep* was also upregulated in *Mesp2*-KO;I-En-KO more than in I-En-KO, but the basal expression of *Anpep* was not affected by the loss of the I-enhancer (Fig 4H). Surprisingly, the inactivation of the P2-enhancer in the I-En-KO background reduced the expression of *Mesp1*, but not *Anpep* (S4C–S4E Fig), indicating that the P2-enhancer activates *Mesp1* in the absence of the I-enhancer. This raised the possibility that the I-enhancer interacts with the P2-enhancer and directs it to act on the *Mesp1* promoter. In summary, the P2-enhancer alone can alter *Mesp1* and *Anpep* expression in the absence of *Mesp2*; however, greater alteration of *Mesp1* expression may require the cooperation of the P2- and I-enhancers (Figs 4G and S4B), supporting case 2 (Fig 4D).

## The P2-enhancer communicates with the promoters of proximal genes at the PSM stage to alter its targets in the absence of the *Mesp2* coding sequence

The P2-enhancer upregulates *Mesp1* and *Anpep* only as a compensatory response (Figs 2D and 4A). We hypothesized that the P2-enhancer interacts with the promoters of proximal genes, *Mesp1* and *Anpep*, when it regulates them. To test this hypothesis, we examined the physical interaction of the P2-enhancer with the promoters of these genes. We employed engineered DNA-binding molecule-mediated chromatin immunoprecipitation (enChIP) [26]. We introduced 3×FLAG-dead Cas9 (dCas9) and gRNA, which recognizes the genomic region close to the P2-enhancer (S5A Fig), into the *iTbx6;iNICD* line and the aforementioned *Mesp2*-CDS-Full-KO line. We hereafter refer to these descendent lines as enChIP lines (S5B Fig). Using enChIP lines, we confirmed the compensatory response in the *in vitro* PSM (S5C Fig). Then, we pulled down the P2-enhancer associated region by FLAG antibody to examine whether the proximal gene promoter regions were associated with the P2-enhancer region in the *Mesp2*-KO condition (See Physical interaction assay by engineered DNA-binding molecule-mediated chromatin immunoprecipitation (enChIP) in Materials and Methods).

We hypothesized that the P2-enhancer can interact with the proximal gene promoters only in *Mesp2*-KO PSM (Fig 5A and 5B). To test our hypothesis, we compared the interactions of the P2-enhancer before and after PSM induction in the control and *Mesp2*-KO conditions (Fig 5C). As expected, we detected the physical interaction of the P2-enhancer with the *Mesp1* and *Anpep* promoters in the PSM, but not in ES cells (Fig 5D). However, these interactions in the KO PSM were comparable with those in the control PSM. This result, contrary to our hypothesis, indicates that the interaction of the P2-enhancer with the proximal gene promoters was established in the PSM, regardless of the presence of *Mesp2*, which negates the assumption of WT genomic conformation in the PSM shown in Fig 5A. The interaction frequency of these gene regulatory regions was not increased in the *Mesp2*-KO line (Fig 5D), indicating that the genomic conformation at the *Mesp2* locus was not significantly different between the control and *Mesp2*-KO cells. Thus, this enhancer-promoter communication was differently utilized between the control and *Mesp2*-KO cells. We speculated that the underlying mechanism of this compensation is the repurposing of this enhancer-promoter communication to activate different targets: the targets of the P2-enhancer are *Mesp2* in the control condition, and *Mesp1* and *Anpep* in the *Mesp2*-KO condition (Fig 6B). The targets of the P2-enhancer may be determined depending on the presence of the *Mesp2* coding sequence (Fig 6B).

To confirm the specificity of these interactions, we examined 60M bp and 17k bp (the same distance as the *Mesp1* promoter oppositely) distant from the P2-enhancer region (NC_60M and NC17k, respectively) (Fig 5C). The P2-enhancer communicated with either the *Mesp1* or

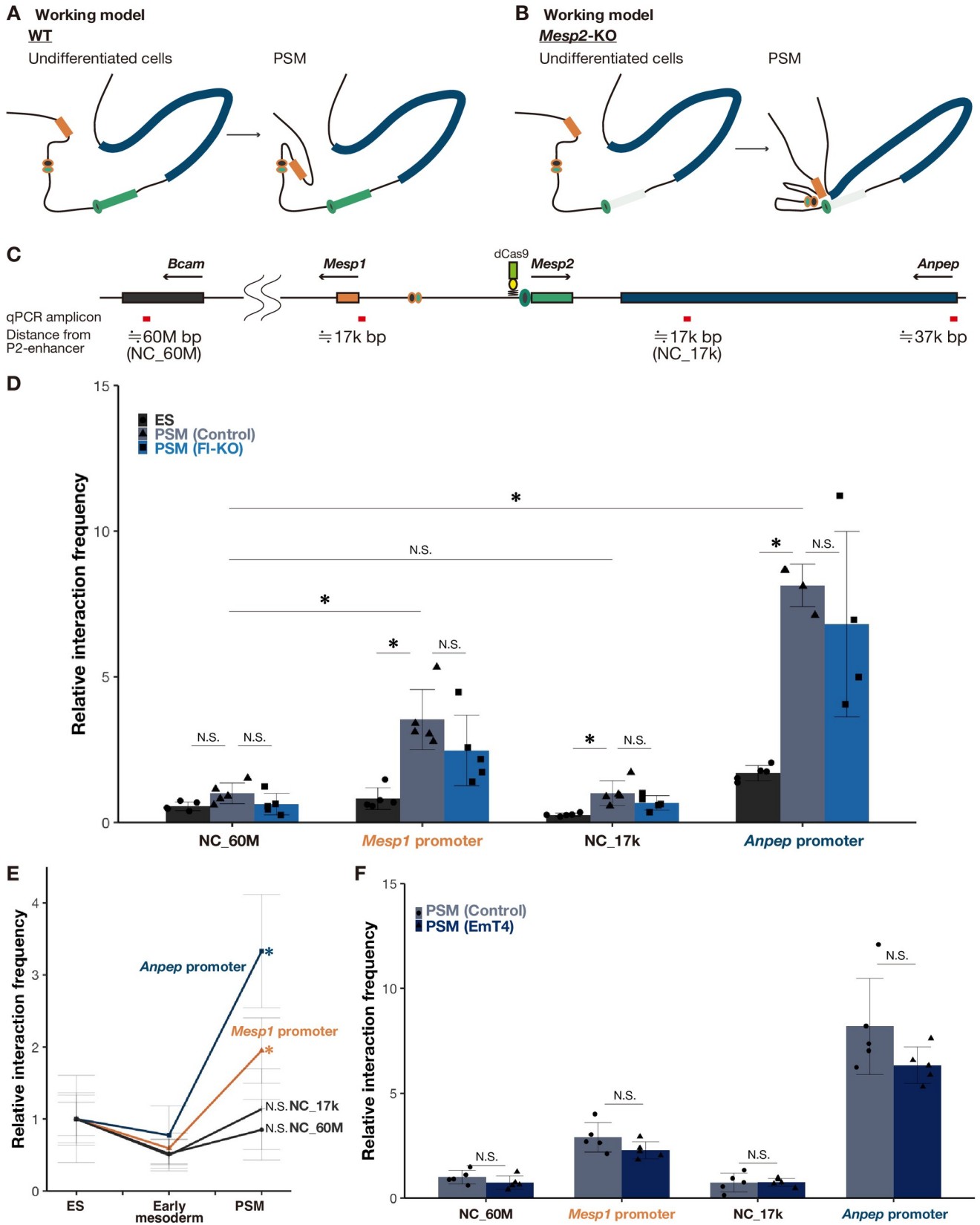

**Fig 5. The communication of the P2-enhancer with proximal promoters was established in *in vitro* PSM independent of *Mesp2* gene disruption.** A, B The genomic conformation models of around the *Mesp2* locus during the differentiation into PSM in the presence (A) and absence (B) of the *Mesp2* gene. Orange, green, and blue rectangles indicate *Mesp1*, *Mesp2*, and *Anpep*, respectively. Circles indicate I- (orange line) and P2- (green line) enhancers. *Mesp2* gene KO is shown as a light green rectangle. C Schematic diagram of the genome around *Mesp2* and *Bcam* regions on chromosome 7. Red bars indicate the examined regions by qPCR for the interaction with the P2-enhancer described in Fig 5. D qPCR analysis of physical interactions of genomic regions in IP input and samples from enChIP cell lines (n = 4 or 5 cultures for each sample). Black, grey, and dark blue bars indicate ES cells (control enChIP cell line) and *in vitro* PSM of control and *Mesp2*-CDS-Full-KO enChIP cell lines, respectively. The Y-axis indicates relative interaction frequency, normalized by the value of the 60M-bp region (NC_60M) from the P2-enhancer in control PSM. *P*-values were calculated by the Mann-Whitney U test comparing regions indicated in this figure. Data are presented as the mean ± SD. *Asterisk* indicates significant ($p < 0.05$). E qPCR analysis of physical interactions of *Mesp1* (Orange) and *Anpep* (Blue) promoters and NC_60M and NC_17k regions described in Fig 5C (Black) in IP input and samples from control enChIP cell lines differentiating from ES cells into PSM (n = 5 cultures for each sample). Early mesoderm indicates the cells treated with BMP medium for two and half days. The Y-axis indicates relative interaction frequency, normalized by the value of each promoter interaction in ES cells. *P*-values were calculated by the Mann-Whitney U test comparing ES cells and others in each promoter and the 60M-bp region (NC_60M) from the P2-enhancer. Data are presented as the mean ± SD. *Asterisk* indicates significant ($p < 0.05$). F qPCR analysis of physical interactions of genomic regions in IP input and samples from enChIP cell lines (n = 5 cultures for each sample). Grey and dark blue bars indicate *in vitro* PSM of control and EmT4 enChIP cell lines, respectively. The Y-axis indicates relative interaction frequency, normalized by the value of the 60M-bp region (NC_60M) from the P2-enhancer in control PSM cells. *P*-values were calculated by the Mann-Whitney U test comparing regions indicated in this figure. Data are presented as the mean ± SD. *Asterisk* indicates significant ($p < 0.05$).

*Anpep* promoter significantly more than these distant regions in the PSM. As these interactions are significantly induced in the *in vitro* PSM compared with in ES cells (Fig 5D), the genomic conformation may develop along with PSM differentiation. Regarding its establishment, we examined these interactions in control cells during the differentiation to PSM. As expected, the interactions were established in the PSM, but not in cells treated with BMP

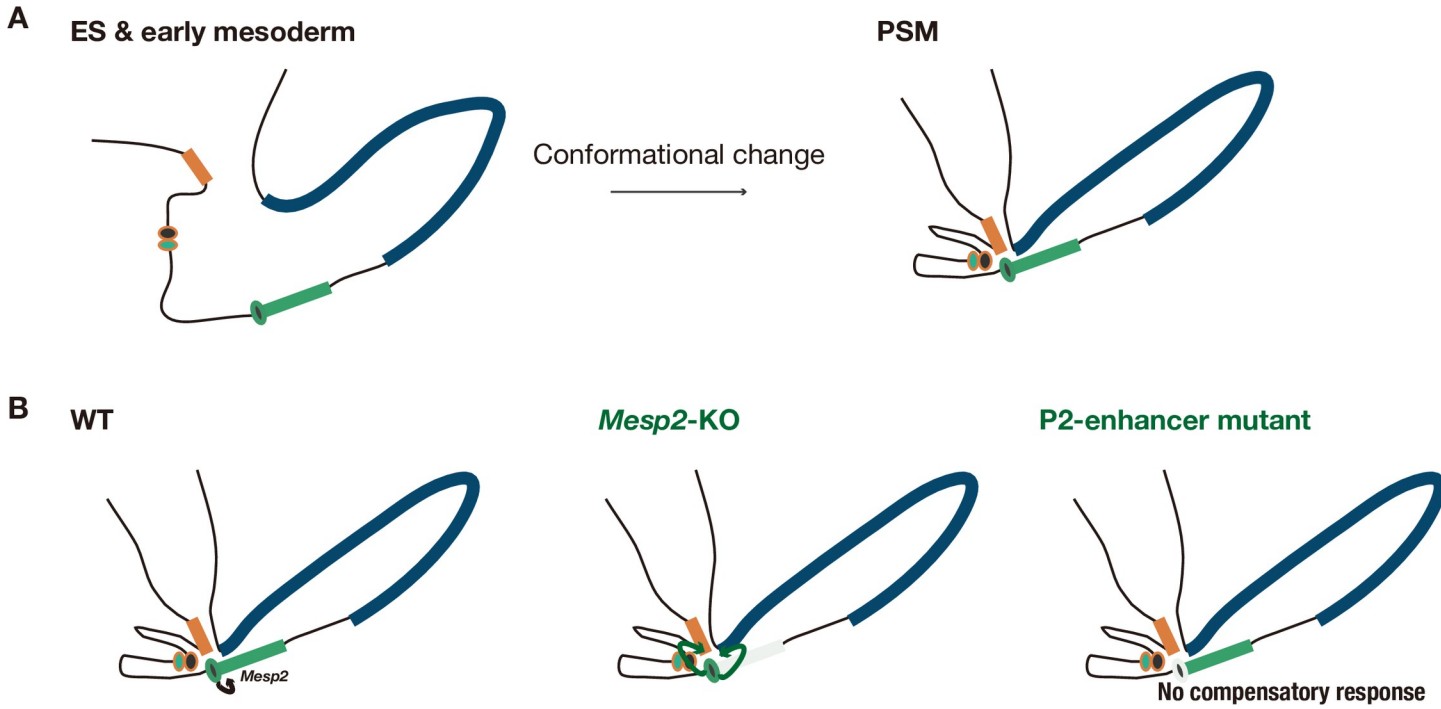

**Fig 6. Proposed genome conformation model of genetic compensation among *Mesp2* and proximal genes.** The model of genomic conformation around the *Mesp2* locus. Orange, green, and blue rectangles indicate *Mesp1*, *Mesp2*, and *Anpep*, respectively. Circles indicate I- (orange line) and P2- (green line) enhancers. *Mesp2* gene KO and P2-enhancer disruption are shown as a light green rectangle and circle, respectively. A In ES cells and early mesoderm cells, the genome around *Mesp2* should be loose (left). In the PSM, the proximal promoters interact with the P2-enhancer (right). B The genomic structure around *Mesp2* is not altered when the *Mesp2* gene is knocked out (shown as a light green rectangle in the middle panel) or the P2-enhancer is inactivated (shown as a light green circle in the right panel), compared with the normal condition (left panel). When the *Mesp2* gene is knocked out, this enhancer-promoter communication is repurposed to activate *Mesp1* and *Anpep*. When the P2-enhancer is mutated, this enhancer no longer functions and the proximal genes are not activated.

medium for two and half days, presumably the early mesodermal cells (Fig 5E). This indicates that the enhancer-promoter communication is established at the PSM differentiation stage (Fig 6A).

The enhancer-promoter communication may play a role in this compensation. Next, we addressed how it is established in the PSM. *Mesp2*, and its upstream regulators, *Tbx6* and *NICD*, were not expressed in ES cells or early mesoderm cells (S1C Fig). The status of the P2-enhancer is active in the PSM but inactive in the ES and early mesoderm cells. The activation status of the P2-enhancer may correspond with the existence of this enhancer-promoter communication (Fig 5D and 5E); therefore, we hypothesized that activation of the P2-enhancer induces this enhancer-promoter communication. To address this possibility, we introduced dCas9 and the same gRNA as above into the EmT4 cell line (Fig 2E), which loses the P2-enhancer activity due to the lack of TBX6 interaction, and investigated the enhancer-promoter communication in this line. However, these interactions did not disappear (Fig 5F). Thus, neither the activation status of the P2-enhancer nor TBX6 binding determines this enhancer-promoter communication. The enhancer-promoter communication remained, but the mutated enhancer was no longer able to activate the proximal gene promoters (Figs 2G and 6B). Although TBX6 is important for *Mesp2*-enhancer activity, this factor was not involved in the establishment of this enhancer-promoter communication, and other regulatory factors are expected.

## Discussion

Genetic compensation is one of the inherent backup mechanisms against unexpected disruption of genes. The underlying mechanism has been ascribed to the induction of downstream signaling in the absence of protein function; however, the molecular mechanisms remain unclear in most cases. Recently, NMD and subsequent pathways were proposed as a mechanism to explain the compensatory response, where PTC-bearing mutant mRNA is involved [4–7]. In this study, we investigated the genetic compensation mechanism of *Mesp* genes, which cannot be explained by the NMD-mediated model, and discovered a novel enhancer-based compensatory mechanism.

The most notable finding related to the enhancer-based compensation mechanism is that the communication of the P2-enhancer with the *Mesp1* promoter is established developmentally in the *in vitro* PSM and repurposed for the compensation event (Fig 6). In contrast to the NMD-mediated model, this mechanism requires the establishment of a genomic interaction to evoke the compensation. How such genomic interactions are established and how enhancer targets are determined are points of interest, as discussed below.

### What defines the genomic conformation to induce enhancer-promoter communication?

In this compensatory event, the enhancer-promoter communication (hereafter called E-P communication) was established during PSM development. The critical question is what mechanism regulates the initiation of E-P communication of *Mesp*. The genome is hierarchically organized and shaped into extruding DNA loop structures, topologically associating domains (TADs), mediated by CTCF and cohesin complex proteins at gene boundaries [27]. Long-range E-P communication, such as crossing TADs, requires domain skipping activity via TAD formation to shorten the inter-domain distance and the distal enhancer to act on the promoter [28]. However, the removal of CTCF or RAD21, a cohesin component, leads to the collapse of TADs, but only modestly affects overall gene expression [29,30]. E-P communication in more local communities was reported to be facilitated by transcriptional components in

some cases [31,32]. We hypothesized that TBX6 induced this communication because this factor determines the activity of the P2-enhancer [14,18]; however, this was not the case (Fig 5F).

TBX6 efficiently binds T-box palindromic sequences as a monomer [33] and also recognizes them as a dimer [14]. TBX21, also known as T-bet, forms a tight dimer and recognizes T-box DNA elements in the promoter and enhancer of *Ifng* simultaneously, then presumably bridges these elements to induce E-P communication at that locus [32]. Contrary to TBX21, other T-box proteins, TBX1, TBX3, and TBX5, recognize the DNA of two T-boxes as a weaker dimer or two monomers. Thus these T-box proteins were assumed to be unable to form E-P communication based on crystal structure analysis [32]. Collectively, TBX6 likely forms a relatively weaker dimer. It may therefore be reasonable to assume that TBX6 is not involved in this E-P communication based on the mode of DNA recognition. Other than proteins of transcriptional components, the transcription at an enhancer, generating noncoding RNA (enhancer-RNA), was reported to establish E-P communication at the immunoglobulin heavy-chain locus (*Igh*) [34]. These studies suggest that E-P communication can be established in diverse ways and a broader view is required for elucidating the responsible factors.

The mode of establishment of E-P communications is categorized into two types, explained by instructive and permissive models [35]. The former model is the *de novo* establishment of an E-P communication triggered by specific stimulation, as described above. The latter is that an E-P communication exists in a preformed 3D conformation that can be used in any cell type by tissue-specific transcription factors for efficient transcription activation, although how the E-P communications are achieved remains unclear [35–40]. The functional difference of E-P communications between preformed and *de novo* formed is unclear [40]. The P2-enhancer case likely matches the permissive model, and it is reasonable to assume that genetic compensation takes advantage of this preformed E-P communication against unexpected gene disruption.

## The enhancer competition theory is one explanation for the repurposing of enhancer-promoter communication

The next critical question is why E-P communication upregulates *Mesp1* and *Anpep* only when the *Mesp2* gene is disrupted, even though the P2-enhancer physically interacts with their promoters in the developed PSM. Such developmentally regulated E-P communications, which do not always correlate with gene activation, were previously reported [39,41,42]. In the *Drosophila* genome, most E-P communications are unchanged between tissue context and across development, and genes are frequently associated with paused RNA polymerase before activation, suggesting that transcription initiates through the release of paused RNA polymerase [39]. In human neuronal cells, E-P communications of essential neural genes are established during the neuronal progenitor phase. Importantly, both up- and down-regulated genes gained E-P communications, suggesting that unknown factors create physical DNA contacts, but do not necessarily cause gene activation [41]. However, which gene among the enhancer communicating genes is selectively activated remains unknown.

The relative distance between an enhancer and promoters was proposed to determine what promoter is preferentially activated by the enhancer [43]. Transcription is not a continuous event, but a bursting event by E-P communication. The proximal promoters compete for enhancer activity, and the closer promoter frequently wins this competition and turns on its transcription [43]. As the P2-enhancer almost overlaps with the *Mesp2* promoter, the *Mesp2* promoter may always win the competition and dominate the *Mesp2*-enhancer. On the other hand, in the absence of the *Mesp2* coding sequence, this enhancer may function as a distal enhancer for *Mesp1* and *Anpep*. This explains why the P2-enhancer is seemingly not involved

in the regulation of *Mesp1* and *Anpep* under normal conditions, and this enhancer-based compensation should be activated upon the loss of *Mesp2*. The switch of an enhancer target without alteration of E-P physical interactions supports this enhancer competition model being applicable to E-P communications that do not accompany gene activation.

The enhancer competition can be interpreted as the competitive target selection by a promoter. However, the molecular mechanisms, especially what determines the enhancer target, remain unclear. Although E-P communication is important for transcription [44], apparent contradictions were also observed. Live imaging of the *Sox2* locus and *Sox2* distal enhancer element (SCR) revealed that although the *Sox2* locus and SCR can get close, *Sox2* is transcribed regardless of the proximity of SCR [45]. As another example, the simultaneous transcription of two genes coregulated by a shared enhancer was observed as two distinct dots separated by 100–200 nm on live imaging, even though their genomic distance was within 3.5k bp [46]. These live imaging experiments suggested that transcription is not simply regulated by physical E-P communication but also by other mechanisms such as transcriptional hubs recruiting RNA polymerase II complexes and associated cofactors. An understanding of general enhancer regulation is necessary to understand the enhancer-based compensation mechanism.

## The mode of the *Mesp*-enhancers as shadow enhancers relevant to genetic compensation

Shadow enhancer is a term first coined by Mike Levine and colleagues in 2008 as an enhancer residing within or on the far side of the neighboring gene, regulating the same gene cooperatively with the proximal enhancer; thus, shadow enhancer implies a redundant and secondary enhancer [47]. The definition of 'shadow enhancer' has been revised because the function of the shadow and proximal enhancers is not clearly different [12,48]. Currently, the definition of 'shadow enhancers' is increasingly accepted as sets of enhancers that regulate a common target gene and drive expression patterns that partially or completely overlap in space and time [12]. The I- and P2-enhancers may be shadow enhancers that regulate *Mesp* genes cooperatively, and this cooperative manner may depend on the enhancers themselves and *Mesp* genes. Most of the target selection by these enhancers can be explained by the "enhancer competition" theory described in the previous section. However, this theory can not explain why the P2-enhancer acts on both *Mesp1* and *Mesp2* only in the absence of the I-enhancer (S4D Fig). The reported shadow enhancer functions can explain this observation and provide further insight into genetic compensation.

There are two possible explanations regarding this enhancer-target selection without the I-enhancer. One is that the I-enhancer inhibits the P2-enhancer action on *Mesp1* under physiological conditions. The other is that the P2-enhancer retargets *Mesp1* for activation after detecting the loss of the I-enhancer. In either case, the I- and P2-enhancers may interact with each other and affect their activities. The modes of shadow enhancer interactions are variable, and they either combine [49–52] or suppress [52–54] their activities to regulate a common target gene, which can be at least partially applied to the mechanism employed by the I- and P2-enhancers. The synergistic upregulation of *Mesp1* by shadow enhancers may be a basis of the genetic compensation, and the repurposing of the preestablished E-P communications may be the same phenomenon observed with shadow enhancers.

## The possible genetic compensation mechanism operating over *Mesp* genes

We started this study to explore a new compensation mechanism for *Mesp* genes independent of the NMD-mediated pathway and revealed enhancer-based compensation. However, in

contrast to the significance of this enhancer-based compensation in our *in vitro* system, P2-enhancer mutant mice, corresponding to EmT3 where no compensation occurs (Fig 2E), exhibited a compensatory response [18]. This strongly suggests another compensation mechanism that functions *in vivo*. What is the difference between *in vivo* and *in vitro*, and what mechanism evokes the compensation *in vivo*? We hypothesize that it is due to the difference in the contribution of TBX6 and the activation of the I-enhancer.

TBX6 is degraded through the RIPPLY2-mediated proteasome pathway at the newly forming somite boundary and is never supplied after somite formation in WT mice. Thus, TBX6 is considered to unbind the I- and P2-enhancers below a certain threshold, which terminates *Mesp* gene expression (S6A Fig), whereas TBX6 likely continues to bind the I-enhancer in *Mesp2*-KO and P2-enhancer mutant mice because of the failure of TBX6 degradation (S6A Fig). Thus, the longer duration of TBX6 binding on the I-enhancer in *Mesp2*-KO and P2-enhancer mutant mice than that in WT increases the I-enhancer activation period and the amount of *Mesp1* transcripts. On the other hand, in our *in vitro* system, TBX6 and NICD were exogenously induced, and the negative feedback loop of TBX6-MESP2-RIPPLY2 did not function properly; thus, the undegraded TBX6 protein level was comparable in control and *Mesp2*-KO cells (Figs 3C and 3D and S6B). Therefore, there is no different occupancy of TBX6 binding on I- and P2-enhancers *in vitro*, leading to no difference in I-enhancer usage.

Of note, the previously reported *Mesp2*-KO mice were generated by replacing *Mesp2* with exogenous genes [9,55]. If this enhancer-based compensation depends on enhancer competition, this compensation mechanism should not be activated in these *Mesp2*-KO mice. Therefore, instead of the enhancer-based compensation, the prolonged activation of the I-enhancer by TBX6 may be the major compensation mechanism operating *in vivo*. In contrast, we propose that only the enhancer-based compensation mechanism functions in *Mesp1* upregulation in our *in vitro* system. We expect that the mutant mice in which P2-enhancer-based compensation occurs will express *Mesp1* more strongly and exhibit clearer rescue, which is required to evaluate the validity of our interpretation.

## The possible impact of this compensation mechanism

The gene order on the chromosome is not random in eukaryotes [56], and adjacent pairs of essential genes for their viability are preferentially conserved through evolution [57]. Consistent with this, the relative genomic arrangement (synteny) of *Mesp1* and *Mesp2* is evolutionarily conserved across zebrafish, mice, and humans. The function and location of the P2-enhancer are also well conserved across mice and fish [18,58]. Moreover, gene pairs that have mutual genomic interactions are highly conserved [59]. Collectively, this implies a conserved genomic interaction and compensatory mechanism at this *Mesp* locus across species.

Gene duplications can supply raw materials for evolution [60,61]. In addition, gene loss can be a pervasive source of genetic change that drives evolution [62,63], including some cases of beneficial effects by gene loss [64]. The prerequisite of the fixation of gene loss in an organism is that the gene is dispensable. The key question is how genes can become dispensable. A study using *Saccharomyces cerevisiae* reported that gene dispensability is predominantly backed up by the transcriptional adaptation of paralogs, known as genetic compensation [65]. Genetic compensation can confer cellular fitness for the survival of an organism; thus, this enhancer-based compensation mechanism can be one of the mechanisms making the gene dispensable. Future studies will reveal homologous gene pairs exhibiting this compensatory response other than *Mesp* genes and elucidate generalities. It may be possible to predict which genes are removed and future evolution by combining genome-wide interaction maps such as Hi-C or ChIA-PET data.

## Materials and methods

### ES cell culture and establishment of modified ES cell lines

ES cells were maintained with feeder cells in ES medium [66]. Several modified ES cell lines were established by transfection using Lipofectamine 2000. To establish the Tet-inducible *Tbx6*-IRES-*NICD* expression system (*iTbx6;iNICD* cell line), a piggyBac transposon system was used as previously described [67]. pBase, CAG-promoter-driven rtTA, and pPB-CMV-*Tbx6*-IRES-*NICD* vectors were transfected into the TT2 ES cells [66]. The ES cell line was selected using neomycin. To generate several *Mesp2* deletion and I-enhancer deletion lines, gRNAs were transiently introduced into the *iTbx6;iNICD* ES cell line. To establish the *iMesp2 or iRipply2* cell lines, a piggyBac transposon system was also used. pBase, PGK-promoter-driven puromycin-resistant gene, and pPB-CMV-*Mesp2* or pPB-CMV-*Ripply2* vectors were transfected into the *iTbx6;iNICD* cell line. To introduce point mutations in the P2-enhancer, a gRNA and template DNA harboring the nucleotide substitution mutations previously described [14,18], in the P2-enhancer, were introduced into the *iTbx6;iNICD* ES cell line. To establish the enChIP lines, a piggyBac transposon system was also used. U6-promoter-driven gRNA that recognizes the DNA region close to the P2-enhancer (shown in S5A Fig) and CAG-promoter-driven 3×FLAG-dCas9 vectors were introduced into *iTbx6;iNICD*, *iTbx6; iNICD;Mesp2*-CDS-Full-KO, and *iTbx6;iNICD;*EmT4 ES cell lines, which were selected using puromycin. The gRNA and dCas9 were continuously expressed in the enChIP lines. All gRNAs were designed using CRISPRdirect [68], the sequences of which are listed in S1 Table.

### *In vitro* PSM induction

For PSM induction, we utilized the method previously described [23] with some modifications. Briefly, the feeder cells were depleted and ES cells were cultured on gelatin–coated culture dishes for two days before inducing PSM differentiation. One or three thousand (only for enChIP experiments) ES cells per well in low-cell-adhesion 96-well plates with U bottoms were first cultured in DMEM/F12 medium supplemented with N2B27 reagent, 1% Knock-out Serum Replacement (KSR), 0.1% bovine serum albumin, 2 mM L-glutamine, 1 mM nonessential amino acids, 1 mM sodium pyruvate, 10 units/ml of penicillin, 10 mg/ml of streptomycin, and 10 ng/ml of BMP4 (BMP medium) for 2.5 days, which was 2 days in the original paper. Cells were then transferred and cultured on human fibronectin-coated dishes with DMEM medium supplemented with 15% KSR, 2 mM L-glutamine, 1 mM nonessential amino acids, 1 mM sodium pyruvate, 10 units/ml of penicillin, 10 mg/ml of streptomycin, 0.5% DMSO, 1 μM CHIRON99021, and 0.1 μM LDN193189 (CL medium) for 2 days. To induce the Tet-inducible gene expression, 1 mg/mL of doxycycline was added into the medium at CL medium day 1 for one day.

### Visualization of protein and RNA

Immunostaining for cultured cells was performed on cover glasses coated with human fibronectin. Cells were fixed by 4%PFA on ice for 10 min, blocked using 3% FBS, followed by incubating with rabbit-anti-TBX6 (1/200) [18] at 4°C overnight and then incubated with Alexa Fluor 488-conjugated anti-rabbit antibody (1/800, Life Technologies, Oregon, USA). For *in situ* hybridization of the mouse *Mesp1* and *Mesp2* mRNA, we used the ViewRNA Cell Plus Assay Kit (Affymetrix, no. TFA-88-19000-99) according to the manufacturer's instructions. Samples were observed using FluoView FV1200 laser scanning confocal microscopy (Olympus).

## Gene expression analysis by real-time quantitative PCR (RT-qPCR)

Gene expression analysis was performed by real-time quantitative PCR (RT-qPCR). Total RNA was isolated from individual samples using TRIzol reagent (Invitrogen) or RNAiso Plus (Takara), treated with DNase I (Invitrogen), and then used for cDNA synthesis with Superscript III or IV (Invitrogen) and oligo-dT primer. RT-qPCR analyses were performed using the Thermal Cycler Dice Real Time System (Takara) or C1000 touch Thermal Cycler and CFX Real-Time PCR Detection System (Bio-Rad) with KAPA SYBR FAST Universal 2X qPCR Master Mix (Kapa Biosystems) with 0.2 µM of specific primers. Cycling conditions were as follows: 95˚C for 3 min; followed by 40 cycles of 95˚C for 10 sec, and 60˚C for 30 sec; and 95˚C for 15 sec, 65˚C for 5 sec and an increase in the temperature to 95˚C to analyze melting curves. Biological replicates are described in figure legends and two technical replicates were conducted. The relative expression levels of genes of interest were calculated by the $2^{-\Delta\Delta Cq}$ method [69]. Cq, ΔCq, and ΔΔCq are quantification cycles, a difference in Cq between samples from a control sample, and the normalized difference of ΔCq by the ΔCq of the internal reference gene, respectively. Gapdh was used as the internal reference gene. Primer sequences are listed in S2 Table, and the specificity of all the primers was confirmed by the single melting peak for each amplicon, and their PCR efficiency was higher than 90%, except for the *Ripply2* primer set, which was 76.7%.

## Western blotting (WB) analysis

The quantification of TBX6 protein was performed by WB. Briefly, the cells were lysed in lysis buffer (150 mM NaCl, 1.0% NP-40, 50 mM Tris-HCl pH 8.0, and cOmplete EDTA-free Protease Inhibitor Cocktail (Sigma-Aldrich)) on ice for one hour, and the supernatants were collected after centrifuging to remove cell debris. Then supernatants were boiled with 2× Laemmli sample buffer (4% SDS, 10% 2-mercaptoethanol, 20% glycerol, 0.004% bromophenol blue, 0.125 M Tris-HCl pH 6.8) for five minutes at 95˚C. Forty micrograms of proteins in the boiled samples was used and separated by SDS-PAGE and then transferred to PVDF membranes (Immobilon). Antibody staining was performed using the following primary antibodies: rabbit anti-TBX6 (1/2000, [18]), or mouse anti-beta-Actin (1/4000, #A5441, Sigma-Aldrich), followed by incubation with goat anti-rabbit IgG conjugated with HRP (1/ 10000, #7074, CST, USA) or donkey anti-mouse IgG conjugated with HRP (1/ 10000, #7076, CST, USA) as the secondary antibodies. Staining with anti-ß-Actin was conducted after mild stripping of the anti-TBX6 staining according to the Abcam protocol (https://www.abcam.co.jp/protocols/western-blot-membrane-stripping-for-restaining-protocol). Protein bands were visualized using SuperSignal West Femto Maximum Sensitivity Substrate (Thermo Fisher Scientific) for TBX6 and SuperSignal West Pico PLUS Chemiluminescent Substrate (Thermo Fisher Scientific) for ß-Actin, and the signals were detected by AE-9300H EZ-CAPTURE MG (ATTO).

## Physical interaction assay by engineered DNA-binding molecule-mediated chromatin immunoprecipitation (enChIP)

To detect physical interactions of the P2-enhancer with the genome regions of interest, we employed engineered DNA-binding molecule-mediated chromatin immunoprecipitation (enChIP). Cells used for enChIP analyses were generated as mentioned in "ES cell culture and modified ES cell lines establishment." The procedure of enChIP analysis was described previously [26]. Briefly, for sample preparation, ES cells 48 hours after seeding $2.0 \times 10^5$ cells without feeders on a gelatin-coated 60mm-dish, and 120 wells of low-cell-adhesion 96-well plates

during the course of PSM differentiation that started with 3000 cells per well were fixed with 1% formaldehyde at 37°C for 5 min. Collected samples were lysed and chromatin fractions were extracted. Chromatin fractions were sonicated and immunoprecipitated by Anti-FLAG M2 Magnetic Beads affinity isolated antibody (Sigma, no. M8823). Fragment DNA without immunoprecipitation (input DNA) and immunoprecipitated fragment DNA were used for the subsequent qPCR analyses described in "Gene expression analysis by real-time quantitative PCR." The occupancy of associated DNA with the P2-enhancer was calculated by dividing the immunoprecipitated fragment DNA amount by the input DNA amount. All the data were normalized by immunoprecipitation efficiency estimated by the pull-down efficiency of the P2-enhancer fragment. Primer sequences are listed in S3 Table.

## Statistical analysis

Data are expressed as the mean ± S.D. The Shapiro-Wilk test was used to assess the normality of distribution of investigated parameters and significant differences were tested using the unpaired two-tailed Mann-Whitney U test. Statistical analyses were performed using the R v.3.6.1 software. Differences were considered significant at $p < 0.05$.

## Supporting information

**S1 Fig. Investigating the expression of developmentally essential genes during PSM development *in vitro*.** A qPCR analysis of *Mesp2*, *Tbx6*, and *Notch1* in *in vitro* PSM using wild-type (WT) ES cells. Failure and Success indicate failed and successful PSM induction using the same WT ES cells (n = 4 cultures for each sample). The expression levels are presented as the ratio against the *in vitro* PSM successfully induced. B Schedule of *in vitro* PSM induction from ES cells to cell harvest to analysis. Dox administration is indicated in red. The duration of Dox administration was 24 hours in every set. Red and black circles are harvest time points with or without Dox administration, respectively. C qPCR analysis of representative developmental genes in *iTbx6;iNICD* during PSM induction with (red bars) or without (grey bars) Dox administration (n = 4 cultures for each sample). The expression levels are presented as the ratio against the *in vitro* PSM cultured in BMP for two and half days and CL media for two days without Dox administration. *Nanog* is a pluripotency marker; *Brachyury* (*T*) and *Eomes* are early mesoderm markers; *Msgn1* and *Hes7* are PSM markers.
(TIF)

**S2 Fig. Summary of the conditions of the *Mesp2* locus and compensatory response.** A Schematic diagram of *Mesp2* gene structure and its deletions or mutations. B Summary of *Mesp1* expression in *Mesp2* mutant lines. The data of *Mesp1* expression are derived from: control, CDS-Full-KO, and Exon1-KO from Fig 1G; P2-Enhancer-KO from Fig 2D; EmT4 from Fig 2G. These data were normalized by *Mesp1* expression in the control in each experiment. *Asterisk* indicates significant ($p < 0.05$).
(TIF)

**S3 Fig. The negative feedback loop does not function properly in the *in vitro* system, probably due to the insufficient induction of RIPPLY2.** A Immunostaining of TBX6 in the *in vitro* PSM of *iTbx6;iNICD* (control), *iTbx6;iNICD;Mesp*2-CDS-KO, *iTbx6;iNICD;Mesp*2-Exon1-KO, and *iTbx6;iNICD;Mesp*2-Enhancer-KO upon Dox administration. Scale bars: 100 μm. B qPCR analysis of *Tbx6* in the *in vitro* PSM (n = 6 cultures for each genotype). The expression level is presented as the ratio against *iTbx6;iNICD* line (control). *P*-values were calculated by the Mann-Whitney U test comparing *iTbx6;iNICD* (control) and *iTbx6;iNICD;Mesp2*-KO lines. Data are presented as the mean ± SD. *Asterisk* indicates significant

(p < 0.05). C Schematic illustration of the generation of *iMesp2* line and *iRipply2* lines. D, E qPCR analysis of *Mesp2* (D) and *Ripply2* (E) in the *in vitro* PSM (n = 6 cultures for each genotype). The expression level is presented as the ratio against *iTbx6;iNICD* line (control). F Immunostaining of TBX6 in the *in vitro* PSM of *iTbx6;iNICD* (control) (left), *iTbx6;iNICD; iMesp2* (middle), and *iTbx6;iNICD;iRipply2* (right) upon Dox administration. Scale bars: 100 μm. G Western blotting analysis to monitor the degradation of TBX6 in the *in vitro* PSM of *iTbx6;iNICD* (control), *iTbx6;iNICD;iMesp2*, and *iTbx6;iNICD;iRipply2* upon Dox administration. ß-Actin was used as the internal control of the protein lysate.
(TIF)

**S4 Fig. The cooperative function of the I-enhancer in this compensation in another genetic background.** A, B qPCR analysis of *Mesp2* (A) and *Mesp1* (B) in *in vitro* PSM; *iTbx6;iNICD* (control), *iTbx6;iNICD*;I-enhancer-KO, *Mesp2*-CDS-Full-KO with the intact I-enhancer, and additive *Mesp2*-CDS-Full-KO on *iTbx6;iNICD*;I-enhancer-KO (n = 5 or 6 cultures for each genotype). The expression levels are presented as the ratio against *iTbx6;iNICD* line (control). *P*-values were calculated by the Mann-Whitney U test comparing *iTbx6;iNICD*;I-enhancer-KO and others. Data are presented as the mean ± SD. *Asterisk* indicates significant ($p < 0.05$). C Schematic illustration of the generation of I-enhancer-KO with additive P2-enhancer mutations (EmT3 or EmT4). D qPCR analysis of *Mesp1* in the *in vitro* PSM; *iTbx6;iNICD* (control), *iTbx6;iNICD*;I-enhancer-KO, and *iTbx6;iNICD*;I-enhancer-KO with additive P2-enhancer mutations (EmT3 or EmT4) (n = 6 cultures for each genotype). The expression levels are presented as the ratio against *iTbx6;iNICD* line (control). *P*-values were calculated by the Mann-Whitney U test comparing samples indicated in this figure. Data are presented as the mean ± SD. *Asterisk* indicates significant ($p < 0.05$).
(TIF)

**S5 Fig. Schematic diagram of enChIP cell lines and the recapitulation of compensation using them.** A Schematic diagram of the gRNA recognition position proximal to the P2-enhancer, which was used for enChIP lines. gRNA sequences are listed in S1 Table. B Schematic illustration of the generation of enChIP lines. C qPCR analysis of *Mesp2* and *Mesp1* in the *in vitro* PSM; *iTbx6;iNICD* (control), and enChIP lines (n = 4 cultures for each genotype). The starting cell number used for control PSM induction per well was 1000. For enChIP cell lines, 3000 cells were also used to induce PSM. The expression of *Mesp* genes in PSM from 1000 or 3000 cells was not different. The expression levels are presented as the ratio against *iTbx6;iNICD* line (control). *P*-values were calculated by the Mann-Whitney U test comparing samples indicated in this figure. Data are presented as the mean ± SD. *Asterisk* indicates significant ($p < 0.05$).
(TIF)

**S6 Fig. Schematic comparison model of the compensation between *in vitro* and *in vivo* *Mesp2* mutants.** Schematic illustration of the course of *Mesp1* and *Mesp2* expression and TBX6 amount in *in vivo* (A) and *in vitro* (B) PSM. Transcripts of *Mesp1* and *Mesp2* are shown as orange and green wavy lines, respectively, and their thickness indicates the strength of transcription. TBX6 protein is shown as a blue T. Binding sites of T on the line indicate I- or P2-enhancer regions. Note that the position of the I-enhancer region is virtually described and not accurate. A Schematic illustration of cells undergoing somitogenesis *in vivo*. TBX6 is gradually degraded by the RIPPLY2-mediated proteasome pathway, and *Mesp1* and *Mesp2* are also gradually degraded in WT. On the other hand, TBX6 remains [18,20,22] and may continue to bind the I-enhancer in *Mesp2*-KO and P2-enhancer mutant mice. The duration of TBX6 binding to the *Mesp1*-enhancer is prolonged, which can increase the amount of *Mesp1* transcripts.

B Schematic illustration of cells after the TBX6 and NICD induction *in vitro*. TBX6 is not fully degraded even in the control due to the exogenous induction of TBX6. The TBX6 binding site on the I-enhancer may be occupied in the control and *Mesp2*-KO conditions. The I-enhancer may always be activated and not differ between the control and *Mesp2*-KO conditions. (TIF)

**S1 Table. Target sequences of gRNA used in this study.** This spreadsheet provides the target sequences of gRNA.
(XLSX)

**S2 Table. List of primers used for qPCR to detect mRNA in this study.** This spreadsheet provides the primer sequences used for cDNA detection by qPCR.
(XLSX)

**S3 Table. List of primers used for qPCR to detect genomic DNA in this study.** This spreadsheet provides the primer sequences used for DNA detection by qPCR.
(XLSX)

**S4 Table. Numerical values for each graph in this study.** This spreadsheet provides the numerical values used for all the graphs.
(XLSX)

## Acknowledgments

We thank Rieko Ajima, Yuzuru Kato, Takamasa Hirano, and Naoko T Fujito (National Institute of Genetics, Japan) for their critical advice and discussions. We thank Makoto Kiso, Masafumi Muraoka, Noriko-Sakurai Yamatani, and Akihiro Maeno (National Institute of Genetics, Japan) for technical support. We thank Danelle Wright (National Institute of Genetics, Japan) for editing this manuscript. We thank Dr. Hitoshi Niwa for providing us piggyBac vectors and Dr. Yukuto Yasuhiko for providing the anti-TBX6 antibody.

## Author Contributions

**Conceptualization:** Hajime Okada, Yumiko Saga.

**Data curation:** Hajime Okada, Yumiko Saga.

**Formal analysis:** Hajime Okada.

**Funding acquisition:** Hajime Okada.

**Investigation:** Hajime Okada.

**Project administration:** Hajime Okada.

**Resources:** Hajime Okada.

**Software:** Hajime Okada.

**Supervision:** Yumiko Saga.

**Validation:** Hajime Okada.

**Visualization:** Hajime Okada.

**Writing – original draft:** Hajime Okada.

**Writing – review & editing:** Hajime Okada, Yumiko Saga.

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
