## [Decision Letter · Decision Letter 0]

18 Aug 2021

Dear Dr Saga,

Thank you very much for submitting your Research Article entitled 'Repurposing of the enhancer-promoter communication underlies the compensation of Mesp2 by Mesp1' to PLOS Genetics.

Your manuscript has been evaluated by three independent peer reviewers. The reviewers all find your work potentially interesting, but also raised important concerns that need to be addressed. Based on the reviews, we will not be able to accept this version of the manuscript, but we would be willing to look at a revised version. We cannot, of course, promise publication at that time.

If you decide to revise the manuscript for further consideration at PLOS Genetics, please aim to resubmit within the next 60 days, unless it will take extra time to address the concerns of the reviewers, in which case we would appreciate an expected resubmission date by email to plosgenetics@plos.org.

[LINK]

Please do not hesitate to contact us if you have any concerns or questions.

Yours sincerely,

Takashi Fukaya

Guest Editor

PLOS Genetics

Wendy Bickmore

Section Editor: Epigenetics

PLOS Genetics

Reviewer's Responses to Questions

**Comments to the Authors:**

Reviewer #1: In the manuscript entitled, “Repurposing of the enhancer-promoter communication underlies the compensation of Mesp2 by Mesp1”, using a cultured presomitic mesoderm (PSM) induction system, Okada and Saga found that the Mesp2-enhancer is required to promote the expression of Mesp1 in response to the loss-function of Mesp2 upon the presomitic mesoderm differentiation. The study provides a compensation mechanism for adjacent genes in the genome, which is different from the nonsense mutation-dependent genetic compensation response. The conclusions are supported by the data. The experiments are well-designed and well described. However, this reviewer has some concerns:

1. The negative feedback loop of TBX6-Mesp2 -Ripply2 in the regulation of Mesp2 and Mesp1 expression suggests that total amount of Mesp1 and Mesp2 is very important for the presomitic mesoderm differentiation. This mechanism is not only for the compensation of loss-function of Mesp2, but also may happen when the expression of Mesp2 is too much. Therefore, it is necessary to determine whether the overexpression of Mesp2 would downregulate the expression of Mesp1 during PSM differentiation.

If it is true, the findings would be more significant. Please the authors discuss this point.

2. Authors should provide more evidence for the negative feedback model in the figure 3. For instance, authors can investigate whether the overexpression of Ropply2 in Ex1-KO and En-KO cells would downregulate the level of TBX6 protein.

3. Using en-ChIP method, authors demonstrated that Mesp2-enhancer physically interacts with Mesp1 and Anpep promoters and Mesp1-enhancer in the PSM but not in ES cells, which is independent of TBX6. The results suggest that topologically associating domains (TADs) have changed from ES cells to the PSM. Authors may find evidence from the 3D map in published data base.

Reviewer #2: The authors have used several genetic manipulations in ES cells and an induced PSM model to determine the potential mechanism of compensation of Mesp genes. The work is well carried out and the results for the most part support the conclusions. The enhancer-based compensation can be inferred from the various deletions of coding vs non-coding parts of Mesp2.

However the enChIP-based results are highly variable and not well controlled as the modest increases in contact frequency could be explained simply by the local chromatin environment in the PSM and not necessarily increased contacts. Most importantly the weak dynamic range of the assay cannot convincingly exclude a change in interaction frequency if this were at play. Indeed no manipulation changes the measured (inferred) interaction frequency. The interaction data should be removed.

The model is confusing as it is inconsistent between Fig 5A and 6A.

Reviewer #3: In this manuscript by Okada and Saga, the authors describe a novel mechanism of genetic compensation – the upregulation of one gene by another’s enhancer in the presence of a gene knock out. In general, the ideas and experimental results are well-described, and the experiments are well-described. The mechanism of genetic compensation is likely to be of interest to many. However, I have several questions and suggestions that relate to linking this observation to previous work and some experimental details that need to be resolved.

Major questions or suggestions:

1. Introduction: There are several pieces of information that would be nice to include in the introduction. I am not an expert in PSM biology, and I had a hard time understanding what the normal expression pattern of Mesp1 is and the phrase starting on line 98. Specifically, the “reciprocal transcription termination of Mesp2” and the “temporal activation and termination of the Mesp2 gene”. Does termination mean repression? If so, I think repression is a better word, since it implies transcription of Mesp2 is blocked, not initiated and then terminated. In addition, the diagram in Figure 3A was helpful – can this or a variant of this be introduced sooner?

2. Identification of enhancers and terminology used to refer to them: At the heart of this paper is the claim that the Mesp2-enhancer can up-regulate Mesp1 in the Mesp2 KO condition. Therefore, much more detail should be provided in to how the Mesp1- and Mesp2-enhancers were identified and what evidence was used to assign them to Mesp1 and Mesp2. Was it simply proximity? Were the enhancers cloned into reporters to determine their expression patterns? In the description, limitations to the previous assignment of enhancers to genes should be discussed.

3. Reference to previous work: There are two related but distinct concepts that this work touches upon. The first is shadow enhancers –redundant enhancers that control a single gene in the same cell type. This term was coined to describe some Drosophila enhancers (https://pubmed.ncbi.nlm.nih.gov/18772429/), but has also been widely discussed in reference to mammalian genes (e.g. https://pubmed.ncbi.nlm.nih.gov/27918583/, reviewed here: https://pubmed.ncbi.nlm.nih.gov/33442000/ and here: https://pubmed.ncbi.nlm.nih.gov/22083793/). This term seems, at a minimum, to apply to the Mesp1 and Mesp2 enhancers in relation to the regulation of Mesp1, but given the data in Figure 4, also to the regulation of Mesp2. The second is the presence of enhancers that are looped to and activate multiple genes. This was described here: https://www.nature.com/articles/nature13417, among other references. This work should both reference and discuss the relationship of the present work to these previous papers.

4. More careful description of the necessity/sufficiency of enhancers: Given the results seen with shadow enhancers, I would suggest being a bit more guarded in the interpretation that the Mesp2-enhancer does not regulate Mesp1 under physiological conditions (line 212, line 381). I think it would be more accurate to say that the Mesp2-enhancer is not necessary under physiological conditions to drive WT levels of Mesp1, but since only a single cell type/time point was assayed, it is possible that the Mesp2-enhancer does regulate Mesp1, even in control conditions, or that it contributes to dynamic aspects of its regulation. Related to this issue, do you think that the non-zero expression of Mesp1 in the P1E-KO is due to the action of the Mesp2-enhancer serving as a backup/shadow enhancer?

5. Description of qPCR results/methodology: Compared to standards like the MIQE ones, the description of the analysis of the qPCR results seems a bit lacking in detail. The units should be specified on all y-axes that show qPCR results. I also think it would make the graphs more interpretable if the y-axis had the same range in all qPCR figures (or at least in panels in one figure). Since results were normalized to Gapdh, it should also be possible to compare the expression of Mesp1 and Mesp2, which may be of interest.

Minor comments:

1. I might choose to call Mesp1 and Mesp2 paralogs instead of homologs, since the term is more specific.

2. Line 58 – “interacted” should be “interacting”

3. Line 75 – the claim is made that NMD is a widely used mechanism of genetic compensation, but no evidence is cited. Has someone actually tried to systematically observe this? Or is this based on the presence of multiple examples? Please clarify.

4. Line 80 – I might call this replacement of the Mesp2 coding region, instead of locus, since locus implies regulatory information was also included (unless it was?)

5. Line 92 – should “domains” actually be “motifs”?

6. Line 113 – To make this more accessible to a broad audience, I might briefly include a description of the published PSM induction system. Otherwise, it is hard to understand the following sentence starting on line 114.

7. Line 118 – Please explain the relationship between the Notch1 and NICD genes.

8. Line 167 – How were the binding sites disrupted? Is there a reason that Mesp1 is more highly expressed in the EmT4 than the EmT3 line?

9. Line 190 – Is there a quantification of TBX6? Representative images seem a bit weak as evidence for the protein level differences.

10. Line 191 – It may be useful to remind the reader of Ripply2’s mechanism of action to explain why protein and mRNA levels of Tbx6 are uncoupled.

11. Line 221 – is Anpep normally expressed in the PSM?

12. Line 231 – “replaced” should be “replacing”

13. Line 316 – I’m not sure “inherent” is the right word to use here

14. Discussion of the regulation of E-P communication: There is more recent work from the Levine lab showing that some E-P distances can be large, even when actively transcribing: https://www.pnas.org/content/116/30/15062. It might be nice to include this citation, though I do agree with the conclusion of this section in that we don’t really know how enhancers chose target promoters.

**Have all data underlying the figures and results presented in the manuscript been provided?**

Reviewer #1: Yes

Reviewer #2: None

Reviewer #3: None

PLOS authors have the option to publish the peer review history of their article (what does this mean?). If published, this will include your full peer review and any attached files.

Reviewer #1: **Yes: **Jun Chen

Reviewer #2: **Yes: **Benoit Bruneau

Reviewer #3: No

---

## [Decision Letter · Decision Letter 1]

6 Dec 2021

Dear Dr Saga,

Thank you very much for submitting your Research Article entitled 'Repurposing of the enhancer-promoter communication underlies the compensation of Mesp2 by Mesp1' to PLOS Genetics.

The manuscript was fully evaluated at the editorial level and by three independent peer reviewers. The reviewers appreciated the attention to an important topic but identified some remaining concerns that we ask you address in a revised manuscript.

We therefore ask you to modify the manuscript according to the review recommendations. Your revisions should address the specific points made by each reviewer.

[LINK]

Yours sincerely,

Takashi Fukaya

Guest Editor

PLOS Genetics

Wendy Bickmore

Section Editor: Epigenetics

PLOS Genetics

Reviewer's Responses to Questions

**Comments to the Authors:**

Reviewer #1: Okada and Saga have extensively revised their MS in response to the reviewer’s concerns. They performed an impressive array of new experiments that strengthened some of their previous claims，there are still some concerns with this manuscript.

(1) In the revised MS, the authors provide new evidence to show that the compensatory response is independent of the TBX6-MESP2-RIPPLY2 negative feedback loop. In addition, a partial deletion in either P1- or P2-enhancer leads to reduction of Mesp1 expression. Thus, it raises a key question for this research, which is whether the compensation response is due to the deletion (or partial deletion) of Mesp2 gene (there is a possibility that this region contains a repressive element), or due to the loss-function of Mesp2 protein. To address this question, the authors might overexpress Mesp2 in the Mesp2-deletion cell lines or knockdown Mesp2 in the control cell line.

(2) If the answer is due to the loss-function of Mesp2 protein, what is the proposed model for loss-function of Mesp2 to upregulate the expression of Mesp1.

Reviewer #2: The authors have carefully addressed my comments.

Reviewer #3: The authors have done a nice job of revising their manuscript. I only have two very small comments:

1. on Line 263 "Anpep expression varied but was not reduced" -- I'd simplify this to "Anpep expression was not significantly reduced"

2. I believe there is a mistake in the x-axis labels of Figure 4F-H -- the last bar is I-En-KO; Ex1-KO, right? P2 wasn't knocked out in these cells, was it?

**Have all data underlying the figures and results presented in the manuscript been provided?**

Reviewer #1: Yes

Reviewer #2: Yes

Reviewer #3: Yes

PLOS authors have the option to publish the peer review history of their article (what does this mean?). If published, this will include your full peer review and any attached files.

Reviewer #1: **Yes: **Jun Chen

Reviewer #2: **Yes: **Benoit Bruneau

Reviewer #3: No

---

## [Decision Letter · Decision Letter 2]

17 Dec 2021

Dear Dr Saga,

We are pleased to inform you that your manuscript entitled "Repurposing of the enhancer-promoter communication underlies the compensation of Mesp2 by Mesp1" has been editorially accepted for publication in PLOS Genetics. Congratulations!

Yours sincerely,

Takashi Fukaya

Guest Editor

PLOS Genetics

Wendy Bickmore

Section Editor: Epigenetics

PLOS Genetics

Comments from the reviewers (if applicable):

Reviewer's Responses to Questions

**Comments to the Authors:**

Reviewer #1: No further questions

**Have all data underlying the figures and results presented in the manuscript been provided?**

Reviewer #1: Yes

PLOS authors have the option to publish the peer review history of their article (what does this mean?). If published, this will include your full peer review and any attached files.

Reviewer #1: **Yes: **Jun Chen

**Data Deposition**

http://datadryad.org/submit?journalID=pgenetics&manu=PGENETICS-D-21-00917R2

**Press Queries**

---

## [Editor Report · Acceptance letter]

11 Jan 2022

PGENETICS-D-21-00917R2 

Repurposing of the enhancer-promoter communication underlies the compensation of </i>Mesp2</i> by </i>Mesp1</i> 

Dear Dr Saga, 

We are pleased to inform you that your manuscript entitled "Repurposing of the enhancer-promoter communication underlies the compensation of </i>Mesp2</i> by </i>Mesp1</i>" has been formally accepted for publication in PLOS Genetics! Your manuscript is now with our production department and you will be notified of the publication date in due course.

With kind regards,

Olena Szabo

PLOS Genetics

On behalf of:
